# Cytoskeletal Protein 4.1G Is Essential for the Primary Ciliogenesis and Osteoblast Differentiation in Bone Formation

**DOI:** 10.3390/ijms23042094

**Published:** 2022-02-14

**Authors:** Masaki Saito, Marina Hirano, Tomohiro Izumi, Yu Mori, Kentaro Ito, Yurika Saitoh, Nobuo Terada, Takeya Sato, Jun Sukegawa

**Affiliations:** 1Department of Molecular Pharmacology, Tohoku University School of Medicine, Sendai 980-8575, Japan; emhirano@gmail.com (M.H.); tomohiro.izumi.p3@dc.tohoku.ac.jp (T.I.); tksato@med.tohoku.ac.jp (T.S.); 2Department of Human Health and Nutrition, Shokei Gakuin University, Natori 981-1295, Japan; jsukegawa@med.tohoku.ac.jp; 3Department of Orthopaedic Surgery, Tohoku University Graduate School of Medicine, Sendai 980-8574, Japan; yu-mori@med.tohoku.ac.jp (Y.M.); itoken_319_rk@yahoo.co.jp (K.I.); 4Center for Medical Education, Teikyo University of Science, Adachi-ku, Tokyo 120-0045, Japan; yurikas@ntu.ac.jp; 5Health Science Division, Department of Medical Sciences, Shinshu University Graduate School of Medicine, Science and Technology, Matsumoto 390-0802, Japan; nobuot@shinshu-u.ac.jp

**Keywords:** cytoskeletal protein 4.1G, primary cilium, preosteoblasts, bone formation

## Abstract

The primary cilium is a hair-like immotile organelle with specific membrane receptors, including the receptor of Hedgehog signaling, smoothened. The cilium organized in preosteoblasts promotes differentiation of the cells into osteoblasts (osteoblast differentiation) by mediating Hedgehog signaling to achieve bone formation. Notably, 4.1G is a plasma membrane-associated cytoskeletal protein that plays essential roles in various tissues, including the peripheral nervous system, testis, and retina. However, its function in the bone remains unexplored. In this study, we identified 4.1G expression in the bone. We found that, in the 4.1G-knockout mice, calcium deposits and primary cilium formation were suppressed in the trabecular bone, which is preosteoblast-rich region of the newborn tibia, indicating that 4.1G is a prerequisite for osteoblast differentiation by organizing the primary cilia in preosteoblasts. Next, we found that the primary cilium was elongated in the differentiating mouse preosteoblast cell line MC3T3-E1, whereas the knockdown of 4.1G suppressed its elongation. Moreover, 4.1G-knockdown suppressed the induction of the cilia-mediated Hedgehog signaling and subsequent osteoblast differentiation. These results demonstrate a new regulatory mechanism of 4.1G in bone formation that promotes the primary ciliogenesis in the differentiating preosteoblasts and induction of cilia-mediated osteoblast differentiation, resulting in bone formation at the newborn stage.

## 1. Introduction

Bone remodeling is coordinated by the balance between bone formation and resorption. Imbalance of the bone formation and resorption causes metabolic bone diseases, including osteoporosis. Disorders of bone formation induce bone brittleness, including osteogenesis imperfecta that is most commonly caused by a defect in collagen. Bone formation is controlled by the activity of osteoblasts. Osteoblasts are differentiated from mesenchymal stem cells (MSCs) through osteoprogenitors and preosteoblasts. Multiple factors are upregulated to promote cell differentiation depending on the differentiation stages. Runt-related transcription factor 2 (Runx2) and osterix are osteogenic transcription factors that are upregulated during the differentiation process and regulate the expression levels of differentiation-associated molecules, including osteocalcin (OC) and alkaline phosphatase (ALP) in preosteoblasts [1,2]. Subsequently, the extracellular matrix is secreted by the osteoblast and located close to the cell surface, and the matrix is mineralized to facilitate osteoblast differentiation.

The primary cilia organized in preosteoblasts play a crucial role in their differentiation to osteoblasts (osteoblast differentiation). The primary cilium is a microtubule-based immotile organelle projecting from the cell surface [3,4]. The cilium regulates cell differentiation and proliferation in various types of cells by mediating signals through selective membrane receptors distributed on its membrane surface, including G protein-coupled receptors (GPCRs), growth factor receptors, and ion channels [5,6,7]. Among these, smoothened, a cilia localizing GPCR, upregulates genes for the transcription factor Gli and Hedgehog (Hh) receptor patched 1 (Ptch1) and is required for bone formation by promoting osteoblast differentiation [8,9]. Furthermore, knockout of cilia formation-associated molecules, including intraflagellar transport (IFT) proteins and kinesins, causes bone malformation, demonstrating the importance of the primary cilia in preosteoblasts for bone formation. For example, single-amino acid deletions in IFT80 cause skeletal manifestations, including a narrow chest, short femur, brachydactyly, and broad hands [10]. Knockout of IFT80, IFT140, and kinesin family member (Kif)-3a in preosteoblasts disrupts the primary ciliogenesis and the cilia-derived Hh signaling, resulting in osteopenia in the femur and tibia [9,11,12]. Likewise, deletion of IFT20 in cranial neural crest (CNC) induces the loss of primary cilia in CNC-derived osteogenic cells and defects in craniofacial skeletal development [13].

Protein 4.1 family is a plasma membrane-associated cytoskeletal protein family that consists of 4.1R (red blood cell type), 4.1G (general type), 4.1N (neuron type), 4.1B (brain type), and 4.1O (ovary type) proteins. This protein family contributes to the stability of the plasma membrane and distribution of membrane proteins at the plasma membrane via interactions with the subcortical actin cytoskeleton and the membrane proteins. Protein 4.1G is widely distributed in a variety of tissues, including the brain, colon, lungs, small intestine, spinal cord, and testis [14]. Studies using 4.1G-knockout mice have revealed multiple roles of 4.1G, including myelin formation in the peripheral nervous system [15,16], maturation of spermatids in the epididymis [17], acquisition of normal visual acuity in the retina [18], and augmentation of cell adhesion, spreading, and migration in mouse embryonic fibroblasts [19]. Furthermore, we previously demonstrated that 4.1G augments and decreases parathyroid hormone receptor (PTHR)-mediated Gq and Gs signaling, respectively [20,21,22]. However, the expression and physiological function of 4.1G in the bone remain unclear.

In this study, we confirmed 4.1G expression in the bone tissue. To investigate the role of 4.1G in bone formation, we used the tibia of a newborn mouse and a mouse preosteoblast cell line, MC3T3-E1, and found that 4.1G is essential for the primary ciliogenesis, the ciliary Hh-signaling, osteoblast differentiation, and the resulting bone formation in the newborn tibia.

## 2. Results

### 2.1. Protein 4.1G Is Essential for Bone Formation in the Newborn Tibia

First, we investigated the expression of 4.1G in bone tissue. For this purpose, we isolated the 4-week-old femurs from male and female mice, removed bone marrow by flushing with phosphate-buffered saline (PBS), minced the bone, and obtained crude cell population that is presumed to include preosteoblasts, osteoblasts, and the rest of bone marrow-derived cells. We confirmed 4.1G expression in the cells by Western blotting, and the expression was abolished in 4.1G-knockout (KO) mice (Figure 1A,B). Next, to explore the role of 4.1G in bone formation, we isolated newborn tibia from male and female mice, followed by a comparison of the area of calcium deposits between WT and 4.1G-KO. The sex of the newborn mice was determined by PCR using genomic DNA. Calcium deposits were observed in the trabecular and cortical bones of the WT and 4.1G-KO tibia (Figure 1C,D). Quantification of the calcium deposit area showed that the overall calcium deposit was reduced in the 4.1G-KO tibia, and the reduction was observed in both males and females (Figure 1E,F). Moreover, the reduction was especially prominent in the trabecular bone of the 4.1G-KO tibia, indicating that 4.1G regulates trabecular bone formation in preosteoblasts and osteoblasts (Appendix A). We also used micro-computed tomography (micro-CT) analysis to measure the tibia diaphyseal diameters because many tibiae were attached with muscle on their surface (Figure 1C,D). The micro-CT imaging showed that the tibia diaphyseal diameters were comparable between WT and 4.1G-KO in both sexes (Appendix A). Here, we could not define the cortical bone thickness and trabecular bone structure by micro-CT analysis because of the very small size of the newborn tibia and the immature tibial structures. Next, we also analyzed the tibial morphology from hematoxylin and eosin sections according to a previous report (Figure 1G,H) [23]. We found that the length of the whole tibia (Figure 1I,J), regions of proliferating and hypertrophic chondrocytes (Figure 1K,L), and region of bone tissue (Figure 1M,N) were comparable between WT and 4.1G-KO tibia in both sexes. These data show that 4.1G is essential for bone formation without affecting bone morphology in the newborn tibia.

### 2.2. Protein 4.1G Is Decreased during the Differentiation of MC3T3-E1 Cells

Because trabecular bone formation was suppressed in the 4.1G-KO tibia, we addressed the molecular mechanism of 4.1G on osteoblast differentiation in the mouse preosteoblast cell line MC3T3-E1. Osteoblast differentiation was promoted in the presence of ascorbic acid and β-glycerol phosphate (AA/βGP) in cell culture systems. We first confirmed the differentiation of MC3T3-E1 cells by analyzing the ALP activity and mineralization of the cells. ALP activity, which started to increase on day 4, was significantly increased 10 days after AA/βGP treatment (Figure 2A). Mineralization of the MC3T3-E1 cell surface, which was detected by alizarin red staining, was increased 12 days after AA/βGP treatment (Figure 2B). Moreover, protein content corresponding to the cell growth was also increased 16 days after AA/βGP treatment (Figure 2C). Interestingly, the 4.1G protein level in the AA/βGP-treated cells was comparable with that in control cells on day 4, but started to decrease on day 8, and progressively decreased from 12 days (Figure 2D,E). However, the mRNA level was not altered by AA/βGP treatment (Appendix A), indicating that the 4.1G protein translation was decreased and/or 4.1G protein was degraded during osteoblast differentiation.

### 2.3. Primary Cilium Is Elongated during the Early Stage of Differentiation in MC3T3-E1 Cells

As shown above, the differentiation and proliferation of MC3T3-E1 cells was augmented by AA/βGP treatment. Because primary cilia could be a cue for both cell differentiation and proliferation, we investigated the relevance of AA/βGP treatment and ciliogenesis in MC3T3-E1 cells. Primary cilia were immunolabeled with an anti-acetylated α-tubulin (Ac-Tub) antibody. We found that Ac-Tub^+^ primary cilia were gradually elongated in a time-dependent manner until day 20 in the control cells, whereas the cilia were rapidly elongated on day 4 in the AA/βGP-treated cells (Figure 3). The ciliary length was still longer in the AA/βGP-treated cells at day 10, but it became comparable between control and AA/βGP-treated cells at day 20 (Figure 3). These data suggest that primary ciliogenesis is accelerated during the early stage (until day 4) of osteoblast differentiation. Considering the result of Figure 1, 4.1G would play important role in the primary cili-ogenesis and osteoblast differentiation, specifically in the early stage. In contrast, it would be less important or even act as a counter-regulator in the primary ciliogenesis, and thus the protein level of 4.1G would be decreased along in the later stage.

### 2.4. Protein 4.1G Promotes Primary Ciliogenesis in the Early Stage of Osteoblast Differentiation

We have shown that 4.1G expression was retained (Figure 2D,E) and the primary cilium was elongated (Figure 3) by the four-days treatment of AA/βGP in MC3T3-E1 cells. Based on these data, we hypothesized that 4.1G contributes to the ciliogenesis in the early stage of osteoblast differentiation prior to the induction of primary cilium-mediated osteoblast differentiation signaling. To test this hypothesis, paraffin sections of the newborn tibia were immunolabeled with an anti-Arl13b antibody, another primary cilium marker, and the number of ciliated cells and the ciliary length were analyzed in the preosteoblast-rich region of the trabecular bone of the newborn tibia. We found that knockout of 4.1G decreased the number of ciliated cells without affecting ciliary length in both males and females (Figure 4A–F). Similarly, knockdown of 4.1G by short hairpin RNAs (shRNAs) of 4.1G (4.1G-sh1 and 4.1G-sh2) in MC3T3-E1 cells resulted in a decrease in the number of ciliated cells under AA/βGP treatment (Figure 4G,H). The knockdown efficiency of 4.1G was confirmed by Western blotting (Appendix A). These data suggest that the presence of 4.1G promotes primary ciliogenesis in the early stages of osteoblast differentiation.

Next, we observed the ciliary/periciliary distribution of 4.1G in MC3T3-E1 cells. Exogenously expressed FLAG-tagged 4.1G (Appendix A) and endogenous 4.1G (Appendix A) were co-immunolabeled with the primary cilium markers Arl13b and Ac-Tub, respectively. Contrary to our expectation, 4.1G did not accumulate at the primary cilium or around the cilium, indicating that 4.1G indirectly regulates primary ciliogenesis in differentiating preosteoblasts. One of the possible mechanisms of ciliogenesis would be the regulation of signal transduction for the ciliogenesis by 4.1G.

### 2.5. Protein 4.1G Accelerates Osteoblast Differentiation by Regulating Ciliary Signaling in Its Early Stage

To address the role of 4.1G in the primary cilium-mediated osteoblast differentiation signaling, we focused on Hh signaling. One of the receptors of Hedgehog signaling smoothened localizes to the primary cilium when the Hh signal is active [24], and the activation of the ciliary smoothened increases mRNA expressions of the transcription factor Gli1 and Hh receptor Ptch1 during osteoblast differentiation [9]. While the treatment of differentiating MC3T3-E1 cells with a smoothened-specific agonist, purmorphamine, for 6, 12, and 24 h resulted in time-dependent increases in Gli1 and Ptch1 mRNA levels (Figure 5A,B; control), knockdown of 4.1G reduced the increased levels of the mRNA (Figure 5A,B; 4.1G-sh1). Purmorphamine also increased the mRNA levels of osterix and osteocalcin in differentiating MC3T3-E1 cells (Figure 5C,D; control), but the increased mRNA levels of both osterix and osteocalcin were decreased by 4.1G-knockdown (Figure 5C,D; 4.1G-sh1). Runx2 mRNA was not increased by purmorphamine treatment, and the mRNA level tended to be decreased by 4.1G-knockdown even though the decrease was not statistically significant (Appendix A). Furthermore, AA/βGP treatment elevated ALP activity on day 4 in the control-transfected cells, but not in the 4.1G-knockdown cells (Figure 6A). In contrast, protein content, which corresponds to cell growth, was not affected by either the AA/βGP treatment or 4.1G-knockdown (Figure 6B). These data suggest that 4.1G augments ciliary osteoblast differentiation signaling in preosteoblasts.

## 3. Discussion

The present study describes a novel mechanism in bone formation: 4.1G promotes primary ciliogenesis in differentiating preosteoblasts, followed by induction of cilium-mediated Hh signaling and the osteoblast differentiation. This mechanism would be responsible for calcium deposition in the trabecular bone. In contrast, 4.1G had no impact on the tibial size, including the length of the whole tibia and proliferating and hypertrophic zones.

We showed that while bone mineralization was well observed in the trabecular and cortical bones of the newborn tibia in WT mice, overall mineralization was reduced in the newborn tibia of the 4.1G-KO mice. In this study, we found that mineralization of the trabecular bones, which was regulated by the actions of peri-trabecular preosteoblasts and osteoblasts, was reduced in the 4.1G-KO tibia. Our results are supported by a previous study, in which deletion of IFT80 in preosteoblasts causes reduction in bone mineralization in the newborn tibia, as well as a decrease in total bone volume, trabecular thickness, and trabecular number and extension of trabecular spacing in 4-week-old mice [9]. In contrast, we showed that 4.1G did not participate in the elongation of the newborn tibia. The comparable length of the proliferating and hypertrophic zones, which accounted for the growth plate size, between WT and 4.1G-KO tibia reflected the results.

This study also demonstrates that 4.1G is essential for primary ciliogenesis, specifically in the early stages of osteoblast differentiation. This observation was supported by two data points. First, the expression level of 4.1G protein was retained in the early stage of differentiation of MC3T3-E1 cells, but it was dramatically decreased in the later stage. Second, primary ciliogenesis was reduced in the newborn 4.1G-KO tibia and in the early stage of differentiation of MC3T3-E1 cells. No study has reported so far that 4.1G is a ciliogenesis- or ciliopathy-associated molecule [25]. To our knowledge, this is the first report to demonstrate the relevance of 4.1G to ciliogenesis. However, 4.1G did not accumulate in the primary cilium and/or around the cilium, indicating that 4.1G indirectly regulates primary ciliogenesis. The number of ciliated cells was increased in vehicle-treated control cells in MC3T3-E1 cells (Figure 4H), indicating the presence of other factors that would coordinately promote the primary ciliogenesis with 4.1G. Protein 4.1G is a subcortical cytoskeletal protein that potentiates signal transduction of membrane receptors [20], so it would augment membrane receptor signaling that promotes ciliogenesis (e.g., non-canonical Hh signal [26]). Here, AA promotes osteoblast maturation and bone formation by influencing the expression of transcription factors and epigenetic regulators [27,28]. We have speculated that AA/βGP treatment increases biogenesis of (1) the ligands in the cells that stimulate the cells in an autocrine manner and/or (2) the receptors that increase the sensitivity to their ligands in the cells. Thus, osteoblast differentiation would not be induced sufficiently in vehicle-treated control cells due to insufficient expression of the ligands and/or the receptors even though primary cilia are formed in the cells. Moreover, the speculation explains the mechanism that knockdown of 4.1G suppresses the primary ciliogenesis, regardless of the presence and absence of AA/βGP. In contrast, although the cell growth was not significantly reduced in 4.1G-KD cells, as shown in Figure 6B, we cannot ignore the possibility that the cell growth was decreased by the knockdown. Thus, it is also important to know whether 4.1G regulates cell proliferation in preosteoblasts and whether cell proliferation promotes ciliogenesis prior to the promotion of osteoblast differentiation in the future. Further studies are required to uncover the mechanisms of primary ciliogenesis regulated by 4.1G in preosteoblasts. Moreover, investigating the relevance of 4.1G to ciliopathy diseases would be an exciting research topic in the future. It is also noteworthy to point out that 4.1G would be unnecessary or would interfere with the maintenance of the cilium structure after the primary cilium was formed, and hence as shown in this report, the 4.1G protein level would be decreased in the later stage of preosteoblast differentiation.

Interestingly, we found that the 4.1G protein was dramatically decreased in differentiating MC3T3-E1 cells in this study. However, the level of 4.1G mRNA did not change, suggesting that translation of the 4.1G protein was decreased and/or degradation of 4.1G protein was accelerated during osteoblast differentiation. A previous report obtained a similar result: Protein 4.1G mRNA was expressed, but 4.1G protein was undetectable in adult mouse kidneys [29]. Additionally, in human erythroblasts, 4.1G protein expression is high at the early stage of differentiation but low in the late stage [30]. The early reports support our idea that 4.1G would play an essential role in primary ciliogenesis and osteoblast differentiation, especially in the early stages.

The cilium length would link with the strength of cilia-mediated signaling. As shown in this study, knockdown of 4.1G decreases the primary ciliogenesis and cilia-mediated signaling. In addition, the elongation of primary cilium promotes the cilia-mediated signaling [31,32,33], whereas the shortening of the cilium reduces the ciliary signaling [34,35]. Thus, we have considered that the primary cilia elongated by AA/βGP treatment promote the cilia-mediated cell proliferation and differentiation signaling during osteoblast differentiation.

The primary cilium controls cell proliferation and differentiation through cilium-localizing membrane receptors. In preosteoblasts/osteoblasts, platelet-derived growth factor receptor alpha, smoothened, bone morphogenetic protein receptor II, and PTHR localize on the primary cilium and are supposed to regulate cell proliferation and/or differentiation [8,13,36,37]. Although we demonstrated the involvement of the smoothened signal pathway in the 4.1G-regulating osteoblast differentiation in this study, 4.1G could also regulate signal transduction mediated through other cilium-localizing receptors. It is interesting to address what ciliary signaling 4.1G regulates in the differentiation of preosteoblasts. As we mentioned before, AA promotes osteoblast maturation and bone formation by influencing the expression of transcription factors and epigenetic regulators [27,28]. It would be interesting therefore to investigate whether AA treatment modulates the expression levels of those cilium-localizing receptors and their ligands in preosteoblasts, and whether 4.1G would have any influence on the signal transduction mediated by those receptors.

Our observation of which 4.1G promotes mineralization of the trabecular bone in the newborn tibia suggests that 4.1G regulates bone mineral density (BMD) and bone formation in childhood. In this study, we showed that 4.1G supports Hh-induced production of osterix mRNA through primary ciliogenesis in MC3T3-E1 cells. Interestingly, reminiscent phenotypes were observed in osterix mutant patients; a frameshift mutation in the *osterix* gene causes recessive osteogenesis imperfecta in a child [38], and single nucleotide mutations in the *osterix* gene result in low BMD in children [39]. Further studies are required to evaluate the relationship between 4.1G and osterix in childhood metabolic bone diseases.

Runx2 is a central regulator of osteoblast differentiation. Although many studies have suggested that Hh signaling upregulates Runx2 expression in MSCs and preosteoblasts [40,41,42], this study demonstrates that purmorphamine treatment failed to increase Runx2 mRNA expression. One of the major differences between the previous studies and our study is the drug-incubation period; the previous studies treated the cells with Hh ligands for more than 5 days, while we treated MC3T3-E1 cells with purmorphamine for up to 2 days because we performed transient knockdown experiments.

In summary, we demonstrated that 4.1G contributes to primary ciliogenesis in differentiating preosteoblast cells, followed by augmentation of cilium-mediated osteoblast differentiation signaling in the cells. Through this mechanism, 4.1G promotes bone formation in the newborn tibia. This study provides a new avenue for understanding the mechanisms of osteoblast differentiation and bone formation.

## 4. Methods

### 4.1. Materials

Fetal bovine serum (FBS) was purchased from PAA Laboratories GmbH (Pasching, Austria). Minimum essential medium, alpha modification (MEMα), mouse anti-glyceraldehyde-3-phosphate dehydrogenase (GAPDH) antibody, L(+)-ascorbic acid (#198-01251), and Alizarin Red S (#011-01192) were purchased from Wako (Osaka, Japan). Rabbit anti-4.1G antibody was purchased from ProteinExpress (Chiba, Japan). Mouse anti-acetylated α-tubulin antibody (Ac-Tub) (clone 6-11B-1, #T6793), mouse anti-FLAG (clone M2, # F1804), β-glycerol phosphate (βGP) (#50020), SIGMAFAST OPD (#P9187), purmorphamine (#SML0868), and EDTA-free protease inhibitor cocktail (EDTA-free) were purchased from Sigma-Aldrich (St. Louis, MO, USA). HistoVT One (#06380-05) was obtained from Nacalai Tesque (Kyoto, Japan). Rabbit anti-Arl13b antibody (#17711-1-AP) was purchased from Proteintech (Rosemont, IL, USA). CF488A-conjugated anti-mouse IgG (#20010), CF568-conjugated anti-mouse IgG (#20105), and CF568-conjugated anti-rabbit IgG (#20098) were purchased from Biotium (Fremont, CA, USA). Horseradish peroxidase (HRP)-conjugated anti-mouse IgG and HRP-conjugated anti-rabbit IgG were purchased from Cell Signaling Technology (Danvers, MA, USA). The Calcium Stain kit (#CVK-1) was obtained from ScyTek Laboratories (West Logan, UT, USA). TRIzol reagent (#15596026) was obtained from Life Technologies (Thermo Fisher Scientific, Carlsbad, CA, USA). PrimeScript RT reagent Kit (#RR037A) and TB Green Premix Ex Taq (#RR820A) were purchased from TaKaRa Bio (Shiga, Japan). Other chemicals were of reagent grade or the highest quality available.

### 4.2. Plasmid Preparation

The targeting sequences of mouse 4.1G-shRNA1 (5′-GGG CAG AGG TTG GGA AAG ACG A-3′) and 4.1G-shRNA2 (5′-GGG ATC CTA CAC CTT GCA GGC C-3′) were designed as described previously [43]. The designed oligonucleotides were synthesized, annealed, and inserted into the pCAG-IRES-GFP vector [43].

### 4.3. Cell Culture

A mouse preosteoblast cell line, MC3T3-E1 (RCB1126), was provided by the RIKEN BRC through the National Bio-Resource Project of the MEXT/AMED, Japan. The cells were cultured in MEMα supplemented with 10% FBS (10% FBS-MEMα). The cells were maintained in a humidified atmosphere at 37 °C and 5% CO_2_. For differentiation, the cells that reached confluence were treated with a differentiation medium (10% FBS-MEMα supplemented with 250 µM ascorbic acid, 50 µM βGP (AA/βGP), and antibiotics) for up to 20 days by replacement with a fresh differentiation medium every two days. Plasmid DNA was transfected into the cells using a Neon transfection system (Life Technologies, Thermo Fisher Scientific) according to the manufacturer’s protocol. The cells were used for experiments two days after transfection.

### 4.4. Mice

The 4.1G-knockout mice were generated as previously described [44]. The mice were housed under specific pathogen-free conditions and were allowed access to a standard diet and water ad libitum. All animal experiments were approved by the Animal Care and Use Committee of Tohoku University. The genotype of the 4.1G gene in mice was determined using specific primers. The sex of the newborn mice was identified by using SX-F and SX-R primers [45]. The primer sequences are described in Appendix A.

### 4.5. Alkaline Phosphatase (ALP) Activity Assay

ALP activity was measured according to a previously reported method with a modification [46]. In brief, MC3T3-E1 cells were differentiated by AA/βGP treatment in a 35 mm dish. The cells were rinsed twice in PBS on the day of the sample harvest. The cells were then lysed with ALP-lysis buffer (PBS supplemented with 1% NP-40 and protease inhibitor cocktail). The lysate was sonicated five times on ice, agitated for 30 min at 4 °C, and centrifuged at 15,000× *g* for 5 min at 4 °C. Equivalent amounts of the obtained supernatant were incubated with 10 mM *p*-nitrophenyl phosphate in ALP assay buffer (50 mM glycine, 0.5 mM MgCl_2_, pH 10.5) at 37 °C. The reaction was terminated by adding 50 mM NaOH, and the absorbance at 405 nm was measured using a microplate reader (Multiskan GO, Thermo Fisher Scientific).

### 4.6. Mineralization

Calcium deposits on differentiated MC3T3-E1 cells were measured as previously reported [47]. In brief, the differentiated MC3T3-E1 cells were fixed in 10% formalin solution for 30 min and incubated in 40 mM Alizarin Red-S solution (pH 4.2) for 45 min at room temperature. After the cells were rinsed twice in phosphate buffer (PB) (pH7.0), the bound Alizarin Red-S was dissolved in PB-containing 10% cetylpyridinium for 60 min at room temperature. The absorbance of the obtained supernatant at 562 nm was measured using a microplate reader (Multiskan GO).

### 4.7. Histology

First, the tail of the newborn mouse was cut to determine both the 4.1G genotype and sex. Next, the newborn tibia was isolated, fixed in 4% paraformaldehyde at 4 °C overnight, embedded in paraffin, and cut into 3.5 µm section. For morphological analysis, the tibia sections were stained with hematoxylin and eosin. The images were obtained using a digital microscope BZ-X800 (Keyence, Osaka, Japan) with a 2 × objective, and the length of the tibial structure was measured using NIH ImageJ software. Calcium deposits on the newborn tibia were visualized by modified von Kossa’s reaction using a Calcium Stain Kit (ScyTek Laboratories). Images were obtained by BZ-X800 using a 2 × objective, and the area of the calcium deposit was quantified using NIH ImageJ software.

### 4.8. Micro-Computed Tomography

Micro-computed tomography (micro-CT) imaging of the tibia of the newborn mouse was performed using a CT scanner for small animals (LaTheta LCT-200; Hitachi-Aloka Medical Ltd., Tokyo, Japan). CT analysis was performed on two mice in male wild-type animals and three mice in the other groups. All procedures were performed according to the manufacturer’s standardized protocols. Samples were scanned over the entire length of the tibia. The slices were of 96 µm thickness, and the voxel size was 96 × 96 × 96 µm^3^. Volume data were obtained for each sample. A multiplanar radial reconstruction was performed using the DICOM display software (ZioCube, Ziosoft, Tokyo, Japan) as previously described [48]. Two transverse diameters that cross at a right angle were measured using the software, and the longest diameter was applied for the quantification.

### 4.9. Protein Extraction and Western Blotting

AA/βGP-treated- or plasmids transfected-MC3T3-E1 cells were lysed in lysis buffer (50 mM Tris-HCl, 150 mM NaCl, 1% Triton X-100, 2 mM EGTA, pH 7.4) containing protease inhibitor cocktail and 1 mM PMSF [22]. The extracted protein samples were mixed with 5 × Laemmli sample buffer (final concentration: 62.5 mM Tris-HCl, 2% SDS, 5% β-mercaptoethanol, 10% glycerol, and 0.5 mg/mL bromophenol blue, pH 6.8). In contrast, the femur was isolated from 4-week-old mice, the bone marrow was washed out using a 26G syringe in PBS, and the obtained bone tissue was minced and suspended in TRIzol reagent. The protein sample in TRIzol reagent was extracted according to the manufacturer’s protocol. The purified protein sample was lysed in 1 × Laemmli sample buffer, separated by SDS-PAGE, and detected by Western blotting. Western blotting, image acquisition, and quantification were described in our previous report [22]. The following antibodies were used: rabbit anti-4.1G (1:2000), mouse anti-GAPDH (1:5000), and HRP-conjugated secondary antibodies (1:10,000).

### 4.10. Immunofluorescence

Differentiated MC3T3-E1 cells were fixed in 4% paraformaldehyde for 10 min at room temperature. Paraffin sections of the newborn tibia were deparaffinized and activated in HistoVT One for 20 min at 95 °C. Immunofluorescence of the cells or sections was performed according to the conventional protocol [49,50]. The following antibodies were used: mouse anti-Ac-Tub (1:1000), rabbit anti-Arl13b (1:1000), mouse anti-FLAG (1:1000), rabbit anti-4.1G (1:100), and fluorescence-conjugated secondary antibodies (1:400). The cell nuclei were labeled by 4′,6-diamidino-2-phenylindole dihydrochloride. Images were acquired using a confocal microscope LSM780 (Zeiss, Oberkochen, Germany) using a 63 × objective with a z-interval of 0.25 µm. The number of primary cilium-positive cells in the tibial section was counted using the confocal images. The ciliary length was measured using NIH ImageJ software. The number of MC3T3-E1 cells with primary cilia was counted under a fluorescent microscope (Zeiss, Axioplan 2 imaging) using a 40 × objective in a double-blind fashion. For counting in transfected cells, only GFP-positive transfected cells were counted (the pCAG-IRES-GFP vector encodes GFP). More than 100 cells were counted for each experiment. Three independent experiments were performed for each condition.

### 4.11. RNA Isolation and RT-qPCR

MC3T3-E1 cells were treated with 2 µM purmorphamine for 6–48 h in the presence of AA/βGP. Total RNA in the cells was isolated using TRIzol reagent, and the total RNA was reverse transcribed using oligo dT primers using the PrimeScript RT reagent Kit. The target sequences of Gli1, Ptch1, osterix, osteocalcin, and Runx2 were amplified at 95 °C for 5 s and 60 °C for 40 s using TB Green Premix Ex Taq reagent with an ABI 7500 Real-time PCR System (Applied Biosystems, Thermo Fisher Scientific, Waltham, MA, USA). The primer sequences are described in Appendix A. The number of complementary DNAs was quantitatively analyzed using ABI 7500 Fast software v2.3 (Applied Biosystems, Thermo Fisher Scientific). Relative mRNA levels were calculated using the 2^−ΔΔCT^ method.

### 4.12. Statistics

Prism v8.0 (GraphPad Software) was used for graph construction and statistical analysis. Data are presented as the mean ± standard deviation (S.D.) or standard error of the mean (S.E.M.), as indicated in the figure legends. Student’s *t*-test and one-way analysis of variance (ANOVA) followed by Tukey’s test (as a post hoc test), or two-way ANOVA followed by Bonferroni’s test (as a post hoc test) were applied to determine the statistically significant differences. Statistical significance was set as *p* < 0.05. The numbers of samples are described in the figure legends.

## Figures and Tables

**Figure 1 ijms-23-02094-f001:**
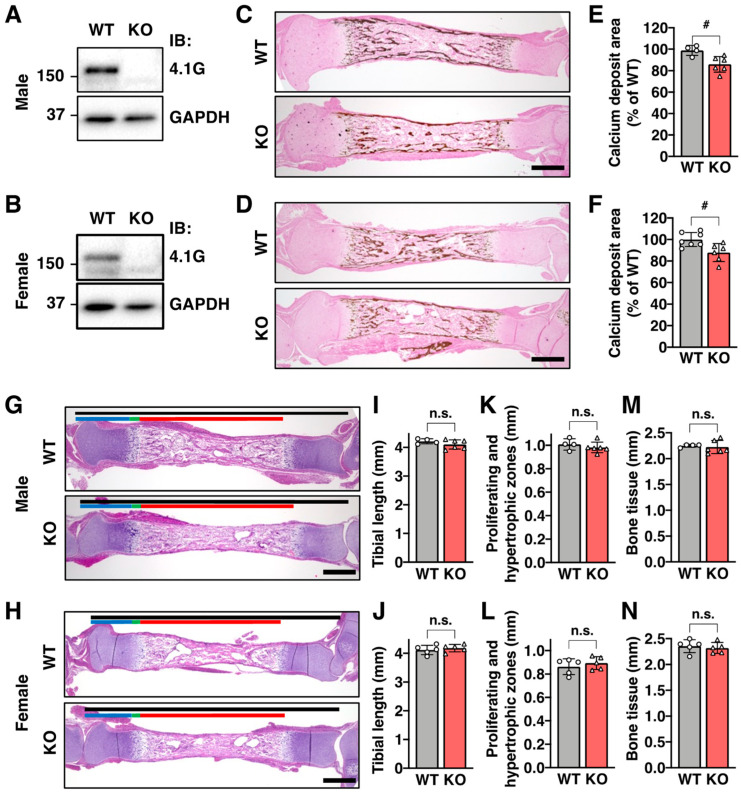
Protein 4.1G regulates osteogenesis of the newborn tibia. (**A**,**B**) Expression levels of 4.1G in wild-type (WT) and 4.1G-knockout (KO) femur. Femurs were isolated from the 4-week-old male (**A**) and female (**B**) mice. Expression levels of 4.1G were analyzed by Western blotting. Cell lysates were immunoblotted (IB) with the anti-4.1G or anti-glyceraldehyde-3-phosphate dehydrogenase (GAPDH) antibody. (**C**–**F**) Calcium deposit. Newborn tibia sections from the male (**C**,**E**) and female (**D**,**F**) mice were stained by modified von Kossa’s reaction. (**E**,**F**) The area of calcium deposit was calculated as %WT of tibia. (**G**–**N**) Tibia morphology. Newborn tibia sections from the male (**G**,**K**,**M**) and female (**H**,**J**,**L**,**N**) mice were stained with hematoxylin and eosin. Black bar, whole tibia; blue bar, proliferating zone; green bar, hypertrophic zone; red bar, bone tissue. The length of the whole tibia (**I**,**J**), proliferating and hypertrophic zones (**K**,**L**), and bone tissue (**M**,**N**). Data are presented as the mean ± standard deviation (S.D.) from 4 ((**E**); WT), 6 ((**E**); KO), 7 ((**F**); WT), 6 ((**F**); KO), 4 ((**I**,**K**,**M**); WT), 6 ((**I**,**K**,**M**); KO), 5 ((**J**,**L**,**N**); WT), and 5 ((**J**,**L**,**N**); KO) independent experiments. # *p* < 0.05, Student’s *t*-test. n.s., not significant. Scale bars, 500 µm.

**Figure 2 ijms-23-02094-f002:**
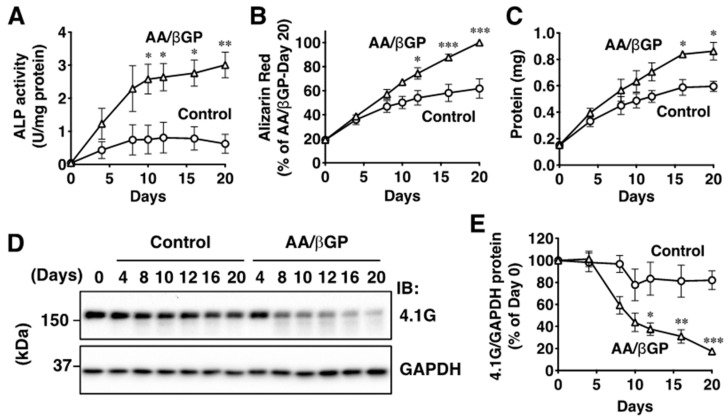
Protein expression levels of 4.1G decrease during the differentiation of MC3T3-E1 cells. The confluent MC3T3-E1 cells were treated with (triangle) or without (circle) 250 µM ascorbic acid and 50 µM β-glycerol phosphate (AA/βGP) for up to 20 days. (**A**) Alkaline phosphatase (ALP) activity in the whole cell lysates was measured at the indicated time points. The activity was normalized by the protein content. (**B**) Calcium deposit on the cell surface was quantified by Alizarin red staining. The intensity of calcium deposit on the AA/βGP-treated cells of day 20 was considered as 100%. (**C**) Protein content in the whole cell lysate was measured at the indicated time points. (**D**) Protein expression levels of 4.1G were analyzed by Western blotting. Whole cell lysates were immunoblotted (IB) with the anti-4.1G or anti-GAPDH antibody. One of four independent representative immunoblots is shown. (**E**) Relative 4.1G expression levels of (**D**). The amount of 4.1G protein expression at day 0 was considered as 100%. Data are presented as the mean ± standard error of the mean (S.E.M.) from 3 (**A**–**C**) and 4 (**E**) independent experiments. * *p* < 0.05, ** *p* < 0.01, *** *p* < 0.001, two-way analysis of variance (ANOVA) followed by Bonferroni’s test (vs. Control of each day).

**Figure 3 ijms-23-02094-f003:**
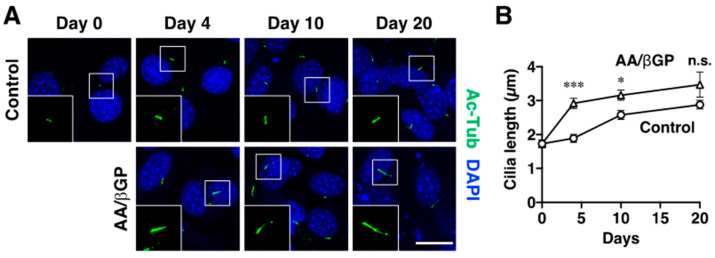
Primary cilia are elongated in the differentiating MC3T3-E1 cells. Confluent MC3T3-E1 cells were treated in the presence or absence of AA/βGP for 4, 10, and 20 days. (**A**) The cells were immunolabeled with an anti-acetylated α-tubulin (Ac-Tub) antibody. The cell nuclei were labeled by 4′,6-diamidino-2-phenylindole dihydrochloride (DAPI). The boxed areas indicate the enlarged regions. (**B**) Quantification of the cilia length. Data are presented as the mean ± S.E.M. from 29–87 cilia. * *p* < 0.05, *** *p* < 0.001, two-way ANOVA followed by Bonferroni’s test (vs. Control of each day). n.s., not significant. Scale bar, 20 µm.

**Figure 4 ijms-23-02094-f004:**
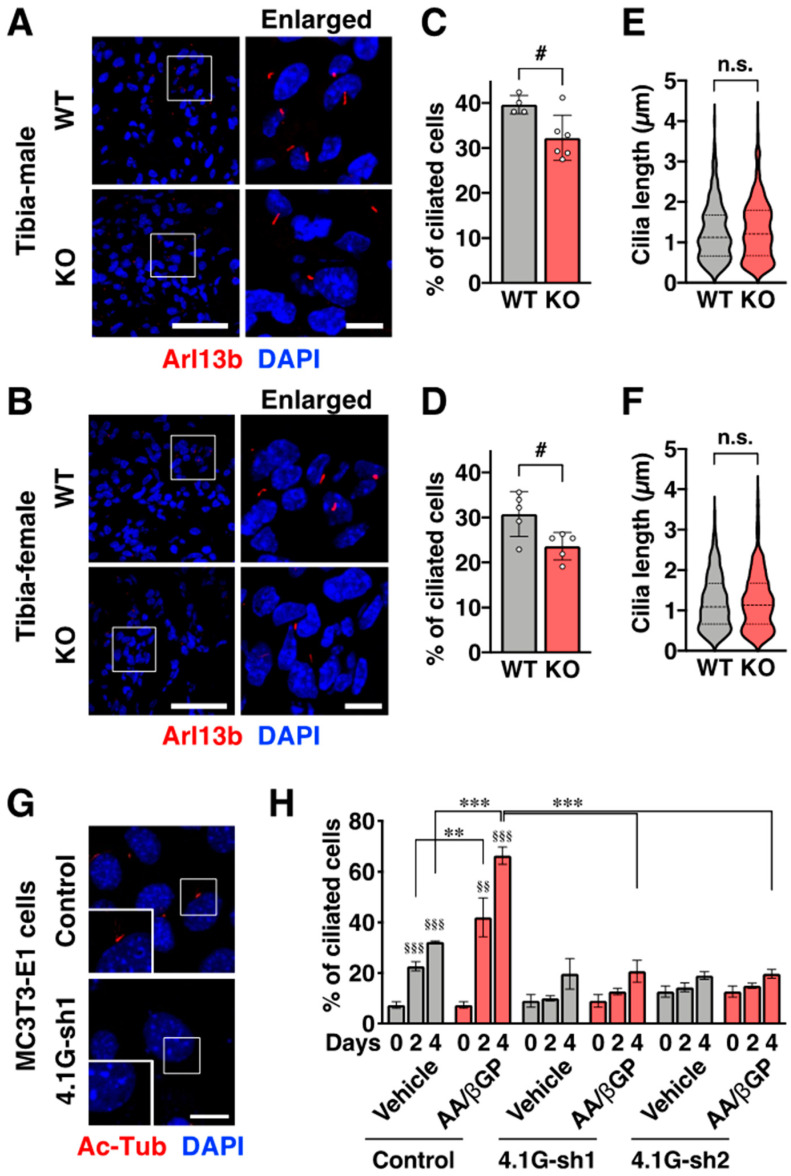
Protein 4.1G promotes ciliogenesis in preosteoblasts. (**A**–**F**) Ciliogenesis at the trabecular bone in newborn tibia. Newborn tibiae were isolated from the male (**A**,**C**,**E**) and female (**B**,**D**,**F**) WT or 4.1G-KO mice. (**A**,**B**) The cilia were immunolabeled with an anti-Arl13b antibody. The cell nuclei were labeled by DAPI. The boxed areas indicate the enlarged regions. (**C**–**F**) Percentages of ciliated cells (**C**,**D**) and cilia length (**E**,**F**) at the trabecular bone are shown. (**G**,**H**) Ciliogenesis in MC3T3-E1 cells. (**G**) MC3T3-E1 cells transfected with the control, 4.1G-sh1, or 4.1G-sh2 were treated with AA/βGP for 2 and 4 days, respectively. Representative confocal microscope images of primary cilia (Ac-Tub) in control- or 4.1G-sh1-transfected MC3T3-E1 cells. (**H**) Percentages of primary cilia-positive cells are shown. (**C**–**F**) Data are presented as the mean ± S.D. from 4 ((**C**); WT), 6 ((**C**); KO), 5 ((**D**); WT), and 5 ((**D**); KO) independent experiments. In the cilia length experiment, 285–380 cilia were measured (**E**,**F**). ^#^ *p* < 0.05, Student’s *t*-test; n.s., not significant. (**H**) Data are presented as the means ± S.E.M. from three independent experiments. ^§§^ *p* < 0.01, ^§§§^ *p* < 0.001, one-way ANOVA followed by Tukey’s test (vs. Day 0 of the group). ** *p* < 0.01, *** *p* < 0.001, two-way ANOVA followed by Bonferroni’s test. Scale bars, (**A**,**B**) 50 µm (wide view) and 10 µm (enlarged view), (**G**) 20 µm.

**Figure 5 ijms-23-02094-f005:**
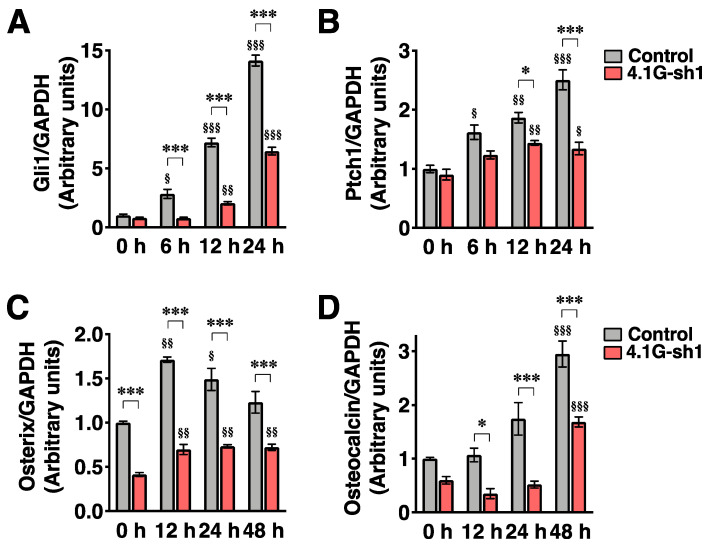
Protein 4.1G increases primary cilia-mediated osteogenic signals. MC3T3-E1 cells transfected with the control or 4.1G-sh1 were treated with AA/βGP for 2 days. The cells were further stimulated with 2 µM of purmorphamine for indicated time periods. Relative mRNA expression levels of Gli1 (**A**), patched 1 (Ptch1) (**B**), osterix (**C**), or osteocalcin (**D**) were normalized to GAPDH level. Data are presented as the mean ± S.E.M. from three independent experiments. ^§^ *p* < 0.05, ^§§^ *p* < 0.01, ^§§§^ *p* < 0.001, one-way ANOVA followed by Tukey’s test (vs. 0 h of the group). * *p* < 0.05, *** *p* < 0.001, two-way ANOVA followed by Bonferroni’s test.

**Figure 6 ijms-23-02094-f006:**
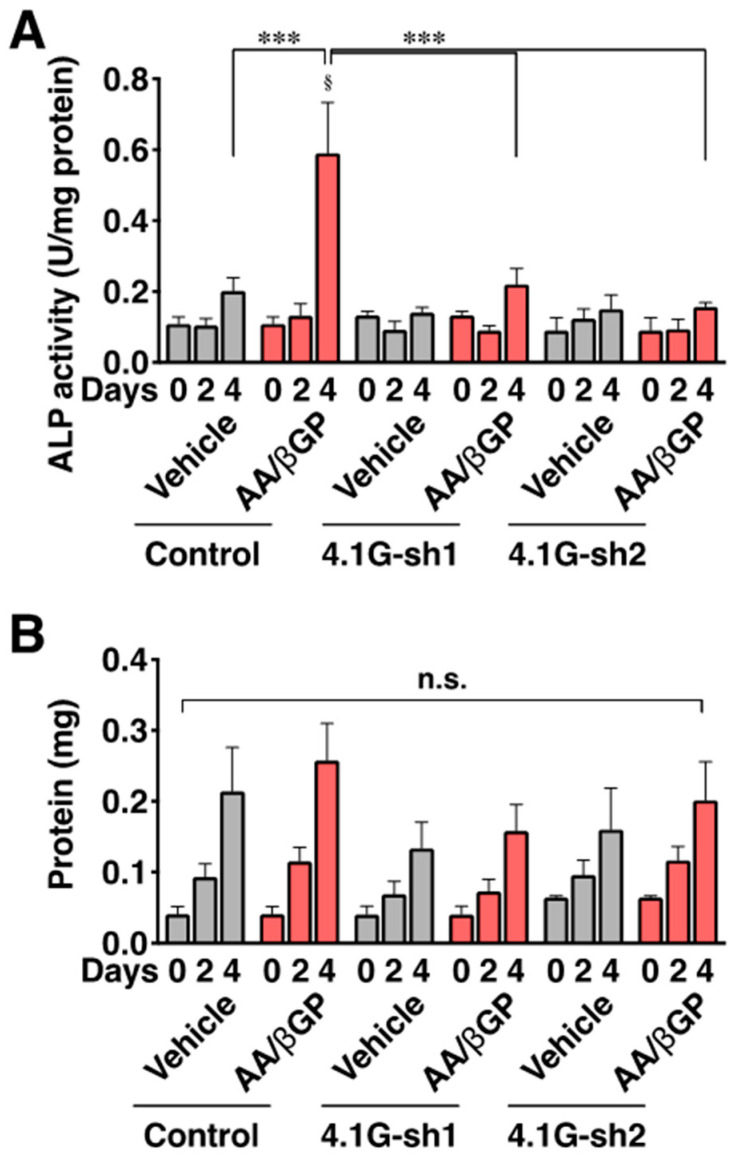
Protein 4.1G supports osteoblast differentiation in MC3T3-E1 cells. MC3T3-E1 cells transfected with the control or 4.1G-sh1 were treated with AA/βGP for 2 and 4 days. (**A**) Alkaline phosphatase (ALP) activity in the cell lysate was measured. The activity was normalized by the protein content. (**B**) Protein content in the whole cell lysate was measured at the indicated time points. Data are presented as the mean ± S.E.M. from five independent experiments. ^§^ *p* < 0.05, one-way ANOVA followed by Tukey’s test (vs. Day 0 of the group). *** *p* < 0.001, two-way ANOVA followed by Bonferroni’s test. n.s., not significant.

## Data Availability

The datasets generated and/or analyzed during the study are available from the corresponding author on reasonable request.

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
