# Peer review of "Cytoskeletal Protein 4.1G Is Essential for the Primary Ciliogenesis and Osteoblast Differentiation in Bone Formation"

_ijms, 2022, doi:10.3390/ijms23042094_

Round 1

Reviewer 1 Report

This is an interesting article revealing a newly disclosed mechanism and inter-relationship between 4.1-G-directed cilliary organization and osteogenic differentiation of osteoblasts in bone formation process.

However, there are some intriguing points that need to be further clarified:

# Figure 1. the author revealed that 4.1G KO may cause the overall reduction of calcium deposition in newborn tibia and claimed that the bony morphology remained unaffected. However, in 1C and 1D, obviously there was a significant discrepancy of tibia diaphyseal diameter among male and female tibia? Could the authors postulate some explanation about this phenomenon? And also the trabecular cortical thickness is markedly reduced in KO group. 

#In Figure 1 legends, the author possibly misused the term" femur" instead of tibia. please make the correction accordingly.

In Figure 2, if 4.1G is so essential for osteoblast differentiation, why osteogenic induction of MC3t3-E1 cells caused a marked reduction of 4.1G protein expression? The results is controversial?

# Figure 3: What is the scientific implications of ciliary elongation?

Overall, i think this article has high scientific merits and deserved to be published after revisions. 

Author Response

Thank you for your comments and suggestions.  Please see the attachment.

Reviewer 2 Report

The authors suggest that 4.1G by affecting the number of cilia modulates osteoblast differentiation. This supposedly takes place through the expression of 4.1G during the first 4 days of MC3T3 in vitro mineralization. There are some conclusions and statements that are incorrect and should be revised. The amount of bone needs to be quantified using a regular evaluation method such as pQCT or mCT. The major issue with the manuscript is that 4.1 G and the change in cilia could be associated but not causally related. In particular, the master regulator of osteoblast differentiation RUNX2 was not affected by the smoothened agonist or by 4.1G knockdown. Furthermore, I don’t understand how a stable amount of 4.1G found in control cells would be associated with an increase initially in cilia numbers that is less pronounced than the cells in differentiating media. The later decrease in length could indeed be due to the drop in 4.1G in differentiating cells, but there is no direct evidence. Based on these points (see also below) it seems the authors need to do more to convince the reader that 4.1G is indeed relevant and causes osteoblast differentiation

Line 40: I disagree. Osteogenesis imperfecta is a defect in collagen and not in the balance. Change.

Line 43: later may differentiate! Not all differentiate to osteocytes.

Line 48: not “of the osteoblast surface” but secreted by the osteoblast and located close to the surface

Line 51: osteoblasts (add the s)

Line 70: this protein family contributes to (change order)

Line 91: “in” change to “with”

I am not convinced that flushing out the bone marrow gives a crude osteoblasts preosteoblast mixture. It gives bone marrow. You need to cut the bones in pieces, let the cells grow out and then collect the cells.

Line 97: it is ridiculous to refer to another paper for how you determined the sex. Just write down how you did that.

Results 2.1. You need to evaluate the bones by another method, such as mCT or pQCT. The amount of trabecular bone depends largely on the section and the differences between CT and KO is small and could therefore be due to the selection of sections.

Line 181: “show” should be changed to “suggest”.

Line 197: remove this sentence: it sounds awkward and does not add anything: to consider the mechanism of the promotion of primary cilia organization by 4.1G,

Line 203: instead of “possibly” write “one possibility would be”

Line 217: it is not shown that mRNA significantly increases. Add p Values. While it can be assumed, it is not shown.

Line 220: Figure 5S: to me it seems that there is a tendency to a decrease in Kd cells that would be seen if the number of replicates was increased.

Line 220: is it possible that ALP would increase later such that the Kd only slows down progression and therefore diminishes the number of cilia?

Line 227: at or for?? Did you treat for 12 hours or did you add it after 12 hours and collect the cells after 2 days? Description is not clear.

Line 229: by GAPDH should read to GAPDH. This is a problem throughout the manuscript but here was confusing.

Line 231: accelerates should be changed to “supports”

Line 242: remove: and the maturation of newborn tibia and change deposit to deposition.

Line 244: remove: and bone tissue.

Line 247-249: that is not accurate. There are also oseoblasts and preosteoblasts in cortical bone vicinity.

Line 253-255: What is bone formation in cartilage?? If you mean in trabecular area I disagree with the conclusion. You did not evaluate the parameters of total bone volume, trabecular thickness, number or spacing in your sections and you should evaluate these three dimensionally as suggested above.

Discussion: 258-262: There is a disconnect: 4.1G is not increased early in MC3T3. It is just still present as it is throughout the time course in control cells. In control cells the number of ciliated cells (the percentage) increases even though these cells are not differentiating or accumulating calcium. This would therefore suggest that the presence or absence is not critical. If anything it would be the number of ciliated cells that modulate differentiation, but this is something you did not address. Please rewrite and stay away from the general statement of “organization”, because it confuses the reader.

Line 297: change to “mediated”

Line 298-304: I don’t see the connection. Just because mutations in osx affect bone and 4.1G presumably (you did not convince me it did) does not mean there is a relationship.

Author Response

(The authors gave the same response as above.)
